

# Effectiveness of Ammonia Reduction on Control of Fine Particle Nitrate

Hongyu Guo[1], Rene Otjes[2], Patrick Schlag[3,4,5], Astrid Kiendler-Scharr[4], Athanasios Nenes[1,6,7,8], Rodney J. Weber[1]

[1] School of Earth and Atmospheric Sciences, Georgia Institute of Technology, Atlanta, GA 30332, USA
[2] Energy Research Centre of the Netherlands (ECN), Petten, Netherlands
[3] Utrecht University, Utrecht, Netherlands
[4] Institute for Energy and Climate Research (IEK-8): Troposphere, Forschungszentrum Jülich, Jülich, Germany
[5] Now at University of Sao Paulo, SP, Brazil
[6] School of Chemical and Biomolecular Engineering, Georgia Institute of Technology, Atlanta, GA 30332, USA
[7] Institute for Chemical Engineering Sciences, Foundation for Research and Technology – Hellas, Patras, GR-26504, Greece
[8] Institute for Environmental Research and Sustainable Development, National Observatory of Athens, P. Penteli, Athens, GR-15236, Greece

* To whom correspondence should be addressed: rweber@eas.gatech.edu

**Abstract.** In some regions, reducing aerosol ammonium nitrate ($NH_4NO_3$) concentrations may substantially improve air quality. This can be accomplished by reductions in precursor emissions, such as nitrogen oxides ($NO_x$) to lower nitric acid ($HNO_3$) that partitions to the aerosol, or reductions in ammonia ($NH_3$) to lower particle pH and keep $HNO_3$ in the gas phase. Using the ISORROPIA-II thermodynamic aerosol model and detailed observational datasets, we explore the sensitivity of aerosol $NH_4NO_3$ to gas phase $NH_3$ and $NO_x$ controls for a number of contrasting locations, including Europe, the US, and China. $NO_x$ control is always effective, whereas the aerosol response to $NH_3$ control is highly nonlinear and only becomes effective at a thermodynamic "sweet spot". The analysis provides a conceptual framework and fundamental evaluation on the relative value of $NO_x$ versus $NH_3$ control. We find that regardless of the locations examined, it is only when ambient particle pH drops below approximately 3 that $NH_3$ reduction leads to an effective response in $PM_{2.5}$ mass. The required amount of $NH_3$ reduction to efficiently decrease $NH_4NO_3$ at different sites is assessed. Owing to the linkage between $NH_3$ emissions and agricultural productivity, substantial $NH_3$ reduction required in some locations



may not be feasible. Finally, controlling $NH_3$ emissions to increase aerosol acidity and evaporate $NH_4NO_3$ will have other effects, beyond reduction of $PM_{2.5}$ $NH_4NO_3$, such as increasing aerosol toxicity and changing the deposition patterns of nitrogen and trace nutrients.




## 1. Introduction

Global trends of increasing gas-phase ammonia ($NH_3$) concentrations (Erisman et al., 2008) have multiple environmental implications. As part of the global nitrogen cycle (Fowler et al., 2013), excessive $NH_3$ deposition promotes alga blooms, degrades water quality, and may be toxic for

ecosystems (Krupa, 2003; Camargo and Alonso, 2006). $NH_3$ is one of the most important atmospheric alkaline species, as it influences the pH of clouds, fogs, precipitation (Wells et al., 1998), and fine particles ($PM_{2.5}$) (Guo et al., 2017c). Agricultural practices, including use of synthetic nitrogen-based fertilizer and domesticated animal manure are the major anthropogenic $NH_3$ sources (Galloway et al., 2003; Aneja et al., 2009; Zhang et al., 2018). Minor contributions

include biomass burning (e.g., forest fires), fossil fuel combustion, and vehicle catalytic converters (Perrino et al., 2002; Behera et al., 2013). Given that fertilizer usage supports food production for about half the global population (Erisman et al., 2008), $NH_3$ emissions are strongly tied to population growth. Compared to the limited regulation of $NH_3$ emissions, emissions of other air pollutants that are linked to acidic atmospheric species, such as sulfur

dioxide ($SO_2$) and nitrogen oxide ($NO_x$), are regulated through air quality standards and account for gas and aerosol concentration decreases observed in the U.S. (Hand et al., 2012; Russell et al., 2012; Hidy et al., 2014), western Europe, and China (Warner et al., 2017). Decreasing trends of $SO_2$ and $NO_x$ emissions are expected to continue on global scales throughout the century (IPCC, 2013). The contrast between increasing $NH_3$ and decreasing $SO_2$ and $NO_x$ leads to changes in

aerosol composition and mass concentration. $NH_3$ reacts rapidly with the oxidized products of $SO_2$ and $NO_x$, sulfuric ($H_2SO_4$) and nitric ($HNO_3$) acids, to form ammonium sulfate (($NH_4$)$_2SO_4$, or other forms such as $NH_4HSO_4$, ($NH_4$)$_3H(SO_4)_2$) and ammonium nitrate ($NH_4NO_3$) aerosols, which globally constitute an important fraction of ambient $PM_{2.5}$ mass (Kanakidou et al., 2005; Sardar et al., 2005; Zhang et al., 2007). These reaction pathways link $NH_3$ to $PM_{2.5}$ mass and its

subsequent impacts on human health (Pope et al., 2004; Lim et al., 2012; Lelieveld et al., 2015; Cohen et al., 2017) and the climate system (Haywood and Boucher, 2000; Bellouin et al., 2011; IPCC, 2013).

A number of studies using regional or global scale models have investigated $NH_3$ controls as a way to reduce $PM_{2.5}$ mass to meet air quality standards (Erisman and Schaap, 2004; Pinder et al.,

2007; Pinder et al., 2008; Paulot and Jacob, 2014; Bauer et al., 2016; Pozzer et al., 2017). The



premise is that reducing $NH_3$ will increase aerosol acidity (i.e., lower aerosol pH) and prevent the formation of $NH_4NO_3$, reducing overall $PM_{2.5}$ mass. Despite the importance of pH, this parameter is often obscured in the regional modeling and not explicitly discussed. Reduction in $NH_3$ also reduces the amount of $NH_4^+$ associated with sulfates, and the interplay between the two

species may drive much of the sensitivity of $PM_{2.5}$ to $NH_3$ and $NO_x$ reductions (e.g., (Vasilakos et al., In review)). Use of large-scale models for this type of assessment requires good predictions of a range of pertinent emissions and sinks ($NH_3$, $NO_x$, $SO_2$, and nonvolatile cations), and accurate representation of applicable atmospheric chemical processes. An important parameter that must be accurately predicted in the model is aerosol pH, which is often done by

an embedded thermodynamic model, such as ISORROPIA-II (Fountoukis and Nenes, 2007). Due to the complexities from all these factors, chemical transport model-predicted responses to changing emissions may not align with observations. For example, the sensitivity of $PM_{2.5}$ pH in the Community Multiscale Air Quality Modeling System (CMAQ) simulations to the mass of crustal material apportioned to the $PM_{2.5}$ size range can have important effects on anticipated

responses to these changing emission trends. Vasilakos et al. (In review) have shown that including too much crustal material in $PM_{2.5}$ results in a predicted increasing trend in both aerosol pH and concentrations of $NH_4NO_3$, which is counter to observations (Weber et al., 2016).

In this study, we use the thermodynamic model ISORROPIA-II directly in a sensitivity analysis to evaluate the effectiveness of $NH_3$ emission controls on fine particle mass relative to $NO_x$

control. Contrasts are made between sites that have a wide range in $NH_3$ concentrations and aerosol composition, with a focus on a one-year dataset collected in Cabauw, Netherlands (Schlag et al., 2016). This site had year around high $NH_3$ concentrations (average $7.3 \pm 6.0$ μg m$^{-3}$, ~ 10 ppbv), with nitrate comprising a significant fraction of the fine particle mass (30% $NO_3^-$ of $PM_1$) and there was a strong seasonal temperature variation. The goal is to establish a

transparent and fundamental understanding on when $NH_3$ emission controls could be an effective way to alter aerosol pH to reduce ammonium nitrate aerosol concentrations, without the use of a full chemical transport model.

## 2. Methods

### 2.1 Sampling sites



*Cabauw:* One-year (July 2012 to June 2013) of online aerosol and gas measurements of inorganic species were made at the Cabauw Experimental Site for Atmospheric Research (CESAR), near the village of Cabauw, Netherlands. Cabauw is a rural site situated approximately 45 km from the Atlantic Ocean (51.970º N, 4.926º E) and surrounded by

agricultural land. With high $NH_3$ concentration, it is somewhat representative of northwestern Europe. Site details, instrumentation, and measurement intercomparisons can be found in Schlag et al. (2016). The data used in this analysis is from a monitor for aerosol and gases (MARGA, Applikon Analytical BV) that was operated by the Energy Research Centre of the Netherlands (ECN). The instrument performs online measurements of soluble inorganic gases collected in a

continuously wetted-wall denuder, followed by a steam-condensation system for collection of particles. Both the aqueous samples of gases and particles are measured via ion chromatography (Schaap et al., 2011; Rumsey et al., 2014), including $NH_3$, $HNO_3$, and $HCl$, and particle phase $NO_3^-$, $SO_4^{2-}$, $Cl^-$, $NH_4^+$, $Na^+$, $K^+$, $Ca^{2+}$, $Mg^{2+}$ alternatively between $PM_1$ and $PM_{2.5}$ at one-hour intervals. Measurement uncertainties were below 10% (Schaap et al., 2011). The detection limits

were 0.05, 0.10, 0.08, and 0.01 µg m$^{-3}$ for aerosol ions $NH_4^+$, $NO_3^-$, $SO_4^{2-}$, and $Cl^-$, respectively, and 0.10 and 0.05 µg m$^{-3}$ for the gases $HNO_3$ and $NH_3$ (Rumsey et al., 2014). Relative humidity (RH) and temperature (T) data were collected at the 2 m level from the CESAR tower and used to represent ground level meteorological conditions (for an overview see Fig. S7 in Schlag et al. (2016)).

*Other Sites*: In addition to the Cabauw site, we analyze the effectiveness of $NH_3$ reduction for a number of contrasting sites where we have already reported on aerosol pH in detail. This includes data from the Southern Oxidant and Aerosol Study (SOAS) (Guo et al., 2015), Wintertime Investigation of Transport, Emissions, and Reactivity (WINTER) (Guo et al., 2016), and California Research at the Nexus of Air Quality and Climate Change (CalNex) study (Guo et

al., 2017a). Briefly, the SOAS data was collected at the Southeastern Aerosol Research and Characterization (SEARCH) Centreville ground site, representative of the southeastern US background conditions, from June to July 2013. The WINTER data was sampled from the National Center for Atmospheric Research (NCAR) C-130 aircraft operating from Feb to March 2015 mainly in the northeastern US. The CalNex data was collected from May to June 2010 in

Pasadena, California, an urban site that is part of the greater Los Angeles region. As a further contrast for regions of very high $NH_3$ concentrations, we include an analysis from published data




in Beijing during winter haze events in 2015 (Wang et al., 2016), for which pH has also been investigated (Guo et al., 2017c). Table S1 summarizes the conditions at the various sites.

## 2.2 Thermodynamic modeling

The thermodynamic model ISORROPIA-II (Fountoukis and Nenes, 2007) was used to determine

the composition and phase state of an $NH_4^+$, $SO_4^{2-}$, $NO_3^-$, $Cl^-$, $Na^+$, $Ca^{2+}$, $K^+$, $Mg^{2+}$, and water inorganic aerosol and its partitioning with corresponding gases. Thermodynamic equilibrium is assumed between fine particles and gases for all semivolatile inorganic species, including particle water and water vapor. Time scales for submicron particles to reach equilibrium are about 30 minutes (Dassios and Pandis, 1999; Cruz et al., 2000; Fountoukis et al., 2009). The

model is run in "forward mode" to calculate gas-particle equilibrium concentrations based on the input of total concentration of inorganic species (e.g., $NH_3 + NH_4^+$, $HNO_3 + NO_3^-$, $SO_4^{2-}$, $Na^+$). $SO_4^{2-}$ has no gas pair as it is virtually nonvolatile in the observed temperature ranges of this study (An et al., 2007). The forward mode gives more accurate and robust results than reverse mode since it is much less sensitive to measurement uncertainties (Hennigan et al., 2015).

Inorganic ions are also assumed to be only in the aqueous phase. This entails a number of assumptions. First, the ambient RH and the history of the particles exposure to RH result in a deliquesced particle. In many cases, diurnal swings in RH (i.e., the maximum RH in early morning) are generally sufficient to reach the deliquescent point. Furthermore, efflorescence RHs are generally low and rarely reached by the ambient RH (10 to 30%) (Bertram et al., 2011).

Thus, a deliquesced particle is often a good assumption when average ambient RH is above 50%. For Cabauw, the one-year mean RH was 81 ± 15 % (± SD), with RH reaching up to 90% during diurnal cycles (see Fig. S1a in the supplement) making the presence of liquid phase a reasonable assumption. For the other sites studied, average RHs were all above 55% (Table S1). A second assumption is that most ions are in an aqueous liquid inorganic phase and only minor fractions

reside dissolved in a separate liquid organic phase, if it exists. This is supported by very good agreement between observed ammonia gas-particle partitioning with thermodynamic model predictions that do not consider an organic phase. (See Figs. S2 and S3 for this study; similar results are found in other studies (Gao et al., 2016; Gao et al., 2017a; Nah et al., 2018). Pye et al. (2018) found only minor difference in predicted ammonia partition when an organic phase was

considered. It is also assumed that the particles were internally mixed, and that pH did not vary



with size. Mixing state of the nonvolatile cations can also affect pH, but the effect on predicted fine particle pH is small if a minor fraction of nonvolatile sulfate is internally mixed with the nonvolatile cations (Guo et al., 2017b), however, it can add uncertainty to predicted nitric acid partitioning (discussed below). Since there is no data on the mixing state and the mass

concentrations (or mole fractions) of nonvolatile cations are generally small (discussed below, see Table S1), internal mixing is assumed in the following analysis.

With increasing pH (e.g., above 2 for oxalate), organic acids can be found at increasing quantities in the particle phase (Nah et al., Submitted). However, organic acids are not considered in the ISORRPIA-II pH calculations. In Cabauw, it has been reported that excess

$NH_4^+$ (i.e., $NH_4^+$ not paired with $SO_4^{2-}$, $NO_3^-$ and $Cl^-$) was correlated with (di-)carboxylic organic acids. Excess $NH_4^+$ on average constituted only 5% of the $NH_4^+$ reported by an aerosol mass spectrometer (AMS) (Schlag et al., 2017) so it is likely to have a small effect on predicted pH. This is confirmed by the good agreement between measured and ISORROPIA-II predicted $NH_3$-$NH_4^+$ partitioning without considering organic acids (see section 3.1). For the winter haze

condition in Beijing, the highest pH among the sampling sites, including organic acids (i.e., oxalate) are reported to decrease pH by at most 0.07, therefore a minor effect (Song et al., 2018).

**2.3 $NO_x$ vs $NH_3$ control to limit PM$_{2.5}$ ammonium nitrate?**

Following the various assessments of $NH_3$ control on PM$_{2.5}$ mass (Erisman and Schaap, 2004; Pinder et al., 2007; Pinder et al., 2008; Paulot and Jacob, 2014; Bauer et al., 2016; Pozzer et al.,

2017), we assume the PM$_{2.5}$ inorganic nitrate is mainly in the form of semivolatile ammonium nitrate and neglect nonvolatile forms, such as $Ca(NO_3)_2$, $NaNO_3$, and similar species, which are generally not found to a large extent in particles smaller than 1 μm. However, it is noted that in locations where concentrations of minerals or sea-salt particle components are high, and the aerosol has aged, formation of semivolatile $NH_4NO_3$ will be perturbed as the $HNO_3$ will evolve

over time to the more stable largely coarse mode salts at the expense of fine mode $NH_4NO_3$ (see Guo et al. (2017a) for example).

Aerosol organic nitrate species can also contribute to aerosol mass (Farmer et al., 2010; Perring et al., 2013; Xu et al., 2015), and may respond to $NO_x$ control, but are not considered here. For the one-year Cabauw data set analyzed here, 9% of the aerosol nitrate was inferred to be organic

nitrate, calculated from the difference in Aerosol Chemical Speciation Monitor (ACSM) nitrate



MARGA-measured nitrate (Schlag et al., 2016). Higher fractions (34% to 44%) have been reported for European submicron aerosols (Kiendler-Scharr et al., 2016). $NO_x$ emission controls could lead to a change in the relative importance of inorganic and organic nitrate (Edwards et al., 2017).

Focusing just on ammonium nitrate, there are two fundamental ways to control $PM_{2.5}$ nitrate; limit the precursors of nitrate aerosol, that is $HNO_3$, or move the nitrate out of the aerosol by reducing the aerosol pH (increasing the particle acidity). The equilibrium aerosol nitrate concentration is given by:

$$NO_3^- = \varepsilon(NO_3^-) \times NO_3^T \qquad (1)$$

where $NO_3^-$ is the concentration in air of semivolatile aerosol nitrate and $\varepsilon(NO_3^-)$ is the fraction
of $NO_3^-$ in the particle phase relative to gas plus particle nitrate ($HNO_3 + NO_3^-$), which is defined as total nitrate, $NO_3^T$. Eq. (1) is the definition of $\varepsilon(NO_3^-)$. Because $\varepsilon(NO_3^-)$ depends on pH, the premise of $NH_3$ control is to reduce $\varepsilon(NO_3^-)$ through decreasing particle pH, whereas $NO_x$ emission controls will mainly reduce $NO_3^T$, although this can also slightly affect pH through aerosol water uptake (discussed below).

**$NO_x$ Control:** Emitted $NO_x$ can undergo a variety of reactions that produce a range of compounds ($NO_z$), including $HNO_3$, peroxynitric acid ($HO_2NO_2$), the nitrate radical ($NO_3$), nitrous acid (HONO), dinitrogen pentoxide ($N_2O_5$), and both gas (e.g., PAN) and particle phase nitrate and organic nitrate species. Once gas phase $HNO_3$ or particle phase $NO_3^-$ is formed, equilibrium between the phases will re-establish gas and particle concentrations. $HNO_3$ is largely
formed by $NO_2$ reaction with the hydroxyl radical (OH), and at night through the nitrate radical-$N_2O_5$ pathway. Modeling studies show that $HNO_3$ can be the most significant of $NO_z$ species (Atkinson, 2000) and is correlated to $NO_x$ emissions (Shah et al., Submitted). Here we assume, to a first approximation, that $NO_x$ mainly produces $HNO_3$ (either directly through reaction with OH or indirectly through production of $N_2O_5$) that partitions to the particle to form semivolatile
aerosol nitrate and rapidly reaches equilibrium. $NO_3^T$ concentrations are then directly related to $NO_x$ control. Use of more detailed modeling approaches can better assess the relationship between $NO_x$ emissions and $NO_3^T$. For example, we are not considering competing chemical pathways that lead to organic nitrates, versus inorganic nitrate that is in equilibrium with gas phase $HNO_3$.





**NH₃ Control:** The effectiveness of ammonia control in reducing $NH_4NO_3$ burdens depends on $\varepsilon(NO_3^-)$ and how it varies with pH, actual pH of the ambient aerosol, and the sensitivity of ambient aerosol pH to changes in $NH_3$ concentration. From thermodynamic equilibrium, $\varepsilon(NO_3^-)$ can be derived from the solubility, reaction (2), and dissociation, reaction (3), of $HNO_3$:

$$HNO_{3(g)} \leftrightarrow HNO_{3(aq)}, \qquad H_{HNO_3} \qquad (2)$$

$$HNO_{3(aq)} \leftrightarrow NO_{3(aq)}^- + H_{(aq)}^+, \quad K_{n1} \qquad (3)$$

Assuming the solution is ideal, $\varepsilon(NO_3^-)$ as a function of pH can be predicted solely based on known properties of $HNO_3$; the $HNO_3$ Henry's constant, $H_{HNO_3}$, and the acid dissociation constant, $K_{n1}$ ($H_{HNO_3}$ and $K_{n1}$ are T dependent), ambient T, and particle liquid water content. The latter is often estimated by only considering water associated with inorganic species ($W_i$; μg m⁻³), determined from measured inorganic aerosol components and relative humidity (RH). Liquid

water associated with organic species can also be included, but normally have minor influence on pH of much lower hygroscopicity and the logarithmic nature of pH (Guo et al., 2015). A more accurate result may be achieved by using measured particle water concentrations.

By combining the equilibrium of reactions (2) and (3):

$$\varepsilon(NO_3^-) = \frac{H_{HNO_3}^* W_i RT (0.987 \times 10^{-14})}{\gamma_{NO_3^-} \gamma_{H^+} 10^{-pH} + H_{HNO_3}^* W_i RT (0.987 \times 10^{-14})} \qquad (4)$$

where 0.987×10⁻¹⁴ is a unit conversion factor, $R$ (J mol⁻¹ K⁻¹) is the gas constant and $H_{HNO_3}^* =$

$H_{HNO_3} K_{n1}$ (mol² kg⁻² atm⁻¹) is the combined molality-based equilibrium constant of $HNO_3$ dissolution and deprotonation, and $\gamma$ are activity coefficients (equal to 1 if assuming an ideal solution). Derivation of Eq. (4) and references for the temperature dependent equilibrium constants, and similar equations for $NH_3$ and HCl partitioning, can be found in the supplemental material of Guo et al. (2017a).

**3. Results and Discussions**

**3.1 The nitrate partitioning S Curve**

The S curve given by Eq. (4) provides a conceptual basis for the effect of ammonia control, through changes in aerosol pH, on particle nitrate. Fig. 1 shows the characteristic "S-shaped"



curve of $\varepsilon(NO_3^-)$ plotted as a function of pH using Eq. (4), for the yearly average conditions in Cabauw and with activity coefficients extracted from ISORROPIA-II ($\gamma_{NO_3^-}\gamma_{H^+} = 0.24$). Including non-ideality shifts the $\varepsilon(NO_3^-)$ S curve to lower pH by approximately 0.6 units.

Fig. 1 shows that there are 3 pertinent pH regions: 1) low pH, where $\varepsilon(NO_3^-)$ asymptotically

approaches 0, and practically all $NO_3^T$ is in the gas phase, 2) $\varepsilon(NO_3^-)$ varies between approximately 0 and 1 and is highly sensitive to pH variations, and, 3) higher pH, where $\varepsilon(NO_3^-)$ approaches 1 and practically all $NO_3^T$ is in the particle phase. This demonstrates that for the one-year average conditions in Cabauw, there is a certain range in ambient pH where $NH_3$ control to alter ambient pH will result in a change in $NO_3^-$ (i.e., region (2) where pH is between 0 and 3).

The greatest change in $NO_3^-$ to a lowering of pH occurs when $\varepsilon(NO_3^-)$ is near 50% (referred to as $pH_{50}$).

It follows that $NH_3$ control will only lead to reduction in $NO_3^-$ if ambient particle pH is within region (2) of Fig. 1. If pH is in region (1) there is no need for $NH_3$ control since pH is sufficiently low that little $NO_3^-$ exists, and if pH is in region (3) the sensitivity of pH to reducing

$NH_3$ will determine the effectiveness of $NH_3$ controls. For example, $NH_3$ first needs to be reduced to move particle pH to the transition point between region (2) and (3), where $\varepsilon(NO_3^-)$ starts to drop. (Note that $NH_3$ control also affects particle mass by changing $NH_4^+$ concentrations, this is discussed more below.)

The S curve of Fig. 1 applies for a given situation (see Eq. (4)), which changes as the particle

composition or ambient conditions (RH, T) change. For example, if $NH_3$ concentrations change, the inorganic particle composition changes, which affects particle water and activity coefficients in Eq. (4), resulting in a shift in the $\varepsilon(NO_3^-)$ curve. Thus, these curves provide only a sense of the general state of how $NO_3^-$ responds to changes in $NH_3$. A full thermodynamic model needs to be run to actually determine the new $\varepsilon(NO_3^-)$ when conditions change. This analysis is provided in

the later part of the paper. The S curve, however, provides valuable insight on sensitivity of $\varepsilon(NO_3^-)$ to pH for a given situation (i.e., what region of Fig. 1).

### 3.2 pH predicted in Cabauw

High concentrations of aerosol inorganic species were observed during the one-year of observations at the CESAR tower. The mass fractions of $NO_3^-$, $SO_4^{2-}$, $NH_4^+$, and $Cl^-$ were on





average 30%, 15%, 14%, and 1%, respectively, of the 9.5 µg m$^{-3}$ particle mass (PM$_1$) (Schlag et al., 2016). The gas-particle partitioning of three semivolatile pairs, NH$_3$-NH$_4^+$, HNO$_3$-NO$_3^-$, HCl-Cl$^-$, measured with MARGA are compared with the thermodynamic model predictions (see section 2 in supplemental material for plots). PM$_1$ and PM$_{2.5}$ MARGA data sets produce similar

results (Fig. S2 versus Fig. S3); here we mainly discuss predictions based on PM$_{2.5}$. Measured and ISORROPIA-predicted partitioning of ammonia was in agreement (NH$_3$: slope = 1.02, R$^2$ = 0.997; NH$_4^+$: slope = 0.97, R$^2$ = 0.96) (Fig. S2). NO$_3^-$ (slope = 1.01, R$^2$ = 0.987) and Cl$^-$ (slope = 0.98, R$^2$ = 0.91) were also in agreement, however, gas-phase components of these two species showed significant discrepancies (R$^2$ of 0.13 to 0.17), possibly due to the gas concentrations

being several times lower than particle concentrations. This can lead to gas denuder measurement uncertainties from particle collection artifacts within the wet denuder. HNO$_3$-NO$_3^-$ and HCl-Cl$^-$ were dominated by particle phases, $\varepsilon$(NO$_3^-$) = NO$_3^-$/NO$_3^T$ = 88 ± 11 % and $\varepsilon$(Cl$^-$) = Cl$^-$/(Cl$^-$ + HCl) = 66 ± 33 %. The opposite was found for NH$_3$-NH$_4^+$, the gas-phase dominated with $\varepsilon$(NH$_4^+$) = NH$_4^+$/NH$_x$ = 19 ± 15 % (total ammonium is referred to NH$_x$ = NH$_3$ + NH$_4^+$),

which is consistent with particle artifacts in the gas collection system possibly affecting HNO$_3$ and HCl, but less effect on NH$_3$. Furthermore, a generally better prediction of NH$_3$-NH$_4^+$ compared to HNO$_3$-NO$_3^-$ and HCl-Cl$^-$ partitioning has been observed in our previous studies and is consistent with the lack of a coarse mode sink for NH$_3$, in contrast to HNO$_3$ and HCl, which can react with sodium and other nonvolatile cations and bias the equilibrium states between fine

particles and gases. In summary, all the semi-volatile inorganic species in the particle-phase (NO$_3^-$, NH$_4^+$, and Cl$^-$) are predicted with high accuracy, as well as NH$_3$-NH$_4^+$ partitioning, therefore, particle water and pH predictions by ISORROPIA-II are expected to be reasonable.

As noted above, the presence of water-soluble nonvolatile cations (NVCs, here include Na$^+$, K$^+$, Ca$^{2+}$, Mg$^{2+}$) can affect the bulk pH analysis. In Cabauw, NVC effects can be assessed by

comparing hourly PM$_1$ and PM$_{2.5}$ data, since these mechanically generated species are largely found in particles larger than 1 µm diameter. Average NVC mole fractions, (i.e., NVCs divided by the total inorganic species, not including liquid water), were consistently small, 5.7% for PM$_1$ and 5.9% for PM$_{2.5}$. However, Na$^+$ was slightly higher in PM$_{2.5}$ at 0.14 ± 0.25 µg m$^{-3}$, compared to 0.05 ± 0.09 µg m$^{-3}$ for PM$_1$. The small and nearly identical fractions of NVCs result in the

same predicted pH for PM$_1$ and PM$_{2.5}$; in both cases pH = 3.7 ± 0.6. Therefore, we focus on the PM$_{2.5}$ in the following discussion due to the similar partitioning predictions and pH for PM$_1$ and





PM$_{2.5}$ (Fig. S2 and S3). A diurnal pattern of ambient particle pH is observed in Cabauw, similar to other studies (Guo et al., 2015), with higher pH of 3.9 at night and lower daytime pH at about 3.5, mainly driven by the diurnal variation in liquid water content (see Fig. S1).

### 3.3 Contrasts in pH and ε(NO$_3^-$) between studies

Fig. 2 includes a comparison of ε(NO$_3^-$) versus pH for the different locations and seasons (Fig. S4 shows separate plots for each region). The ε(NO$_3^-$) curves are plotted based on the campaign average conditions (i.e., T, $W_i$, and $\gamma_{NO_3}-\gamma_{H^+}$; all listed in Table S1). Two sub data sets in Cabauw, summer (June-Aug 2012) and winter (Dec 2012-Feb 2013), are shown together with the one-year whole data set. As seen for Cabauw, lower temperatures (dark blue vs. red vs.

orange lines in Fig. 2) shifts HNO$_3$-NO$_3^-$ partitioning to favor the particle phase due to effect of T on nitric acid Henry's law and dissociation constants, and the explicit effect of T in Eq. 4. For example, at given activity coefficients and liquid water levels, a decrease from 20 °C (~summer) to 0 °C (~winter) shifts ε(NO$_3^-$) to lower pH by roughly one unit. The differences between the ε(NO$_3^-$) curves are also caused by variations in liquid water, and to a lesser degree by variation in

activity coefficients. In general, the summer curves (the right three curves) are at higher pH and the winter curves are at lower pH.

In addition to the S curves, Fig. 2 shows the average ambient particle pH predicted by ISORROPIA-II for each of the studies. Note that pH could also be inferred from the S curve and measured ε(NO$_3^-$) but is more uncertain and requires activity coefficients for non-ideality effects.

A comparison between Eq. (4)-predicted ε(NO$_3^-$) versus pH and observed ε(NO$_3^-$) versus ISORROPIA-II predicted pH is shown in Fig. S5 and confirms consistency between the ISORROPIA-predicted pH and S curve given by Eq. (4). (A plot of ε(NH$_4^+$) vs pH is also shown in Fig. S5). Fine ambient particle pH varies amongst the sites. The pH of 3.7 ± 0.6 in Cabauw is higher than several other regions, such as the SE US (pH = 0.9 ± 0.6), the NE US (0.8 ± 1.0), and

the SW US (1.9 ± 0.5), but slightly lower than the Beijing haze ambient particle pH of 4.2. The higher ambient particle pH is generally associated with higher concentrations of NH$_3$ and NO$_3^-$. Particle pH is affected by coupling between many variables, hence the need for a thermodynamic model. ISORROPIA-II predicts the overall resulting equilibrium values and associated pH. Particle nitrate has a secondary effect on pH by increasing particle liquid water and diluting H$^+$

aqueous concentrations, resulting in slightly higher pH. This effect is less pronounced when





$SO_4^{2-}$ levels exceed $NO_3^-$, meaning that liquid water is mainly controlled by nonvolatile $SO_4^{2-}$. Thus, $NH_3$, $NO_3^-$, and particle pH are coupled. Regions of higher $NH_3$ will have higher pH which can lead to higher $NO_3^-$ (when in Region (2) of Fig. 1). The highest observed $NH_3$ (12.8 μg m$^{-3}$) and $NO_3^-$ (26 μg m$^{-3}$) concentrations were found for the Beijing haze condition. The Cabauw

one-year average $NH_3$ was lower at 7.3 μg m$^{-3}$, and $NO_3^-$ was on average of 4.7 μg m$^{-3}$. The lowest $NH_3$ and $NO_3^-$ levels were observed in the US studies. For example, 1.37 μg m$^{-3}$ $NH_3$ and 3.58 μg m$^{-3}$ $NO_3^-$ in the SW US, and only 0.39 μg m$^{-3}$ $NH_3$ and 0.08 μg m$^{-3}$ $NO_3^-$ in the SE US, both in summer.

The intersection of the $\varepsilon(NO_3^-)$ S curves with ambient particle pH in Fig. 2 (i.e., intersection of

vertical line and corresponding site S curve), provide contrast in the average $\varepsilon(NO_3^-)$ at each site, and hence if and how much $NH_3$ control will be needed to shift $\varepsilon(NO_3^-)$ to 50% and corresponding pH of pH$_{50}$. The lowest $\varepsilon(NO_3^-)$ was found in the SE US at 22% in summer and a higher $\varepsilon(NO_3^-)$ in the NE US in winter at 39%. The Cabauw site also had higher $\varepsilon(NO_3^-)$ in winter (91%) than summer (84%). Additionally, the SW US site observed on average 54%

$\varepsilon(NO_3^-)$ in summer and China haze in winter had ~100% $\varepsilon(NO_3^-)$. These data show that in the eastern US in summer, $\varepsilon(NO_3^-)$ is generally so low that shifting pH by changing $NH_3$ emissions will not greatly influence $NH_4NO_3$ concentrations since most is already in the gas phase. Higher $NH_3$ can increase $NH_4NO_3$, but large changes in $NH_3$ are needed in these regions to change pH (Weber et al., 2016). For the SW US summer, $NO_3^-$ partitioning is sensitive to changes in pH

with $\varepsilon(NO_3^-)$ 54%. In Beijing winter, substantial decrease in pH is needed to evaporate $NH_4NO_3$, even more so than Cabauw in winter. For Cabauw, a substantial reduction in ambient pH would be needed to evaporate $NO_3^-$ since the current pH is on the flat zone of the S curve (Region 3), where $\varepsilon(NO_3^-)$ is near 100%. In summer, however, a much smaller reduction in ambient particle pH would result in a decrease in $NO_3^-$.

**3.4 Simulation of particle mass reduction with a thermodynamic model**

*3.4.1 Sensitivities of pH and nitrate partitioning to NH$_3$ concentration*

In the above analysis, $\varepsilon(NO_3^-)$ versus pH curves relative to ambient particle pH are used to provide insight on how $\varepsilon(NO_3^-)$ is expected to change with small changes in pH. The S curves are based on the average ambient conditions for each time period, and variables, such as particle





water and activity coefficients are held constant. But changes in $NH_3$ concentration will vary aerosol composition, liquid water content and the activity coefficients, which in turn modulates the S curve, Eq. (4). To address this, in the following analysis, we run ISORROPIA-II for various input $NH_x$ concentrations, while T, RH, $NO_3^T$ and $SO_4^{2-}$ are held constant, and plot

various parameters of interest. This takes into account the various aerosol composition and gas phase species concentrations through considering the partitioning of all semi-volatile species, including water, and how this affects thermodynamic properties, such as activity coefficients.

First, we consider the extent of $NH_3$ control needed to reduce $NH_4NO_3$, which depends on the response of pH to changes in ambient $NH_3$ concentration, which in turn is related to $NH_3$

emissions (i.e., changes in $NH_x$). In a previous study, we show that for average conditions at the various sites discussed above, a general rule is that an order of magnitude reduction in $NH_3$ lowers pH by about one unit (Guo et al., 2017c) ($\Delta pH/\Delta(\log_{10}NH_3)$, are listed in Table S1). At the Cabauw site, the responses in pH to changes in $NH_3$ are similar to these other locations; the linear fitted curves for the semi-log plot in Fig. 3a give slopes of 1.00 in winter, 1.16 in summer

and 1.05 for the one-year average (all $R^2 > 0.99$). Fig. 3a also shows predicted pH versus measured $NH_3$ based on hourly average data. How pH changes with temperature for a constant $NH_3$ can also be seen in Fig. 3a; higher temperature leads to lower particle pH due to volatilization of semivolatile $NH_4^+$, $NO_3^-$, and particle water. The physical explanation for this is that with higher temperature, $NH_4^+$ is converted to $NH_3$ and releases one $H^+$ to the particle phase,

whereas $NO_3^-$ is converted to $HNO_3$ and results in loss of one $H^+$ from the particle phase. The former process dominates over the latter due to the differences in temperature dependency of equilibrium constants (see Fig. S6) and the greater loss of $NH_4^+$ from $NH_4NO_3$ and $(NH_4)_2SO_4$ compared to less loss of $NO_3^-$ only from $NH_4NO_3$, leading to a net increase in particle $H^+$ and lower pH. The loss of water associated with $NH_4^+$ and $NO_3^-$ further reduces pH, as the $H^+$

becomes more concentrated. The water effect is also seen in the diurnal pH trends (see Fig. S1b).

This analysis also permits assessing how $\varepsilon(NO_3^-)$, the sum of $NH_4^+$ and $NO_3^-$ ($NH_4^+ + NO_3^-$), and $\varepsilon(NH_4^+)$ responds to changes in $NH_3$. Fig. 3b shows that it takes a factor of 1000 change in $NH_3$ concentration (~3 pH units) to reduce $\varepsilon(NO_3^-)$ from ~100% to ~0% (i.e. from complete particle-phase to complete gas-phase). Also, a change temperature of ~8 °C shifts $\varepsilon(NO_3^-)$ equivalent to

roughly an order of magnitude change in $NH_3$ concentration. (For reference, $\Delta T$ between winter and one-year averages is 7.6 °C and $\Delta T$ between one-year average and summer averages is



8.8°C). Fig. 3b & 3c again show that larger reductions in $NH_3$ are needed in winter compared to summer to reduce $NO_3^-$. In Cabauw, only during the highest temperature periods is a $NH_3$ control policy immediately effective.

Finally, the response of $\varepsilon(NH_4^+)$ to changes in $NH_3$ is shown in Fig. 3d. The S curves are

reversed compared to $\varepsilon(NO_3^-)$ due to opposite base and acid partitioning responses to changes in pH. Thus, lowering $NH_3$ reduces $\varepsilon(NO_3^-)$, reducing $NO_3^-$ for constant $NO_3^T$, but raises $\varepsilon(NH_4^+)$ as the particles become more acidic, resulting in relatively more $NH_4^+$ in the particle phase and less $NH_3$ in the gas phase. This is important since although we discuss $NH_3$ emissions, changes in particle pH also affects $NH_3$ concentrations through changes in gas-particle partitioning, (i.e.,

$\varepsilon(NH_4^+)$), but it is $NH_x$ that is really changing through emission controls.

Finally, Fig. 3d shows that temperature has little effect on the $\varepsilon(NH_4^+)$ versus $NH_3$ curves. This is because for constant $W_i$ and activity coefficients, the $\varepsilon(NH_4^+)$ versus pH S curves move in the opposite direction with change in temperature than the $\varepsilon(NO_3^-)$ versus pH S curves; $\varepsilon(NH_4^+)$ shifts to a lower pH region and $\varepsilon(NO_3^-)$ shifts to a higher pH region with increasing temperature.

This tends to bring the $NH_3$-$NH_4^+$ partitioning versus $NH_3$ curves together and separate the $HNO_3$-$NO_3^-$ partitioning versus $NH_3$ curves for different seasons (Fig. 3c).

### 3.4.2 Effects of $NH_3$, $NO_x$, and $SO_2$ emission control in Cabauw

Here we assess the relative merits of $NH_3$, $NO_x$, and $SO_2$ control on various aspects of $PM_{2.5}$ in Cabauw, again using the full thermodynamic model. Changes in pH, particle water ($W_i$), $\varepsilon(NO_3^-)$,

mass of $NH_4^+ + NO_3^-$, and overall $PM_{2.5}$ ion mass are assessed when changes are made to $NH_x$ ($NH_3 + NH_4^+$), $NO_3^T$ ($HNO_3 + NO_3^-$), and $SO_4^{2-}$, representing control of $NH_3$, $NO_x$, and $SO_2$ emissions, respectively. Each are reduced in steps starting from 0% to a 90% reduction, while holding the other model inputs constant. The results are shown in Fig. 4. The base values are the one-year, summer, and winter average conditions and correspond to 0% reduction in all plots.

The first row in Fig. 4 shows that all parameters respond nonlinearly to $NH_x$ reduction, remaining relatively constant until ~70% $NH_x$ reduction, at which point they start to rapidly decrease. This is a result of the $\varepsilon(NO_3^-)$ versus pH S curve of Fig. 1, where little effect is realized until pH reaches a critical value of about 3 (the horizontal dash line in Figs. 4a, 4b and 4c pH plots). Once pH drops below this, the balance between $HNO_3$ and $NO_3^-$ is sharply shifted





towards the gas phase due to the combined effects of reduced particle pH and also reduced particle water ($W_i$). An approximate 70% reduction in $NH_x$ is required in Cabauw, in winter or based on the yearly average data, to achieve effective reductions in ($NH_4^+ + NO_3^-$) and particle ion mass. In summer, some minor reductions in the mass concentrations occur for small $NH_x$

reductions, since pH is slightly lower in summer (3.3) compared to winter (3.9). Despite the seasonal variations in gas and particle composition, RH and T, all three pH curves (one-year, summer, winter) appear to be similar and show a critical pH of approximate 3; $NH_x$ reduction is more effective for pH below 3 but far less effective for pH above 3, consistent with the simplified analysis above (see Fig. 1).

Effects of reducing $NO_3^T$ (the 2$^{nd}$ row, Fig. 4b, i.e., $NO_x$ control) and $SO_4^{2-}$ (the 3$^{rd}$ row, i.e., $SO_2$ control) show different responses. For $NO_x$ control, holding $NH_x$ and $SO_4^{2-}$ constant, a linear reduction in $NO_3^T$ causes a linear decrease in $W_i$, ($NH_4^+ + NO_3^-$) and $PM_{2.5}$ ion concentrations simply because $\varepsilon(NO_3^-)$ remains close to 1 so that $NO_3^- \sim NO_3^T$. Then a reduction $NO_3^T$ is just transmitted directly to $W_i$ ($SO_4^{2-}$ is constant so particle hygroscopicity is controlled by $NO_3^-$),

($NH_4^+ + NO_3^-$) and $PM_{2.5}$ ions. $\varepsilon(NO_3^-)$ is relatively constant (more so in winter) because it is ~100% and so not sensitive to the changes in $W_i$. Lower $W_i$ does shift the $HNO_3$-$NO_3^-$ S curve towards a higher pH, but since pH is affected little, and never drops below the critical value of 3, $HNO_3$-$NO_3^-$ partitioning is barely affected by reducing $NO_3^T$ (i.e., remains in Region (3) in Fig. 1)

In the case of $SO_4^{2-}$ reduction, particle pH only increases slightly with substantial $SO_4^{2-}$ reduction

due to buffering by $NH_3$-$NH_4^+$ partitioning (i.e., $NH_4^+$ volatility) (Weber et al., 2016; Guo et al., 2017c). ($NH_4^+ + NO_3^-$) decreases slightly due to the loss of associated $NH_4^+$ due to both the drop in $SO_4^{2-}$ and volatilization caused by reduced particle water. Since $SO_4^{2-}$ is nonvolatile and no gas-particle partitioning is involved, the $SO_4^{2-}$ reduction results in a linear reduction in particle ionic mass, while model input of $NH_x$ and $NO_3^T$ are constant.

Sensitivity test were also performed to investigate the robustness of these results. Considering the observed decreasing trends of $SO_2$ emissions in many regions (Hand et al., 2012; Hidy et al., 2014; Warner et al., 2017), we tested a cleaner future with less sulfate (20% of the current level, see Fig. S7 in the supplement). Also, since significant changes in global climate and surface land cover can result in a dustier future with more NVCs, we investigated the effect of a 400%

increase in NVCs above the Cabauw levels (see Fig. S8). These two assumed scenarios produce





a similar conclusion as the base simulation discussed above, including our finding of a critical pH of 3 and nonlinear response to a $NH_x$ reduction. We do note, however, that in the reduced $SO_4^{2-}$ case, $SO_4^{2-}$ control had nearly no effect on particle ion mass because of the very low $SO_4^{2-}$ concentrations to begin with in the cleaner future scenario.

In summary, the optimal strategy to reduce ammonium nitrate or particle total inorganic ion mass for the current conditions in Cabauw is to control $NO_3^T$ ($NO_x$ emission) since it results in a linear response. Even $SO_4^{2-}$ control is superior over $NH_x$ control to reduce particle ion mass, unless over 70% reduction in $NH_x$ could be achieved. If $NH_x$ is reduced, the effects will be greatest in warmer periods. These are also the times when $NH_3$ emission are largest both in Cabauw (Table

S1) and in other regions of generally high $NH_3$ concentrations, such as Asia (Zhang et al., 2018), and so there may be other benefits to controlling $NH_3$ emissions at these times, for example, minimizing eutrophication in surface aqueous systems.

The above findings in Cabauw are in contrast to results of a global model, which also utilized ISORROPIA-II (Pozzer et al., 2017). They find the impacts of $NH_3$ emissions on $PM_{2.5}$ mass is

strongest in winter for Europe (along with North America, and Asia). Some of the differences are likely attributed to our higher predicted pH in Cabauw of ~3.7 compared to the average pH of Europe predicted in the global model to be near 2 (Pozzer et al., 2017). Thus, we predict conditions above the critical pH of 3, and Pozzer et al. (2017) predicts pH below this value. Difference in pH may be due to meteorological conditions or the concentration of aerosol and

gas inorganic species, but it does demonstrate the sensitivity of responses to what the local ambient pH is, and that care should be taken to evaluate predicted particle pH against inferences from ambient measurements. Next, we explore the outcomes of $NH_x$ reductions in other locations and show that $NH_3$ emission control is more effective in winter than summer.

### 3.4.3 Effects of NH₃, NOₓ, and SO₂ emission control for other locations

$NH_x$, $NO_3^T$, and $SO_4^{2-}$ reduction tests were also run for the other sampling sites following the same approach as described above for Cabauw. The model input (period averages) can be found in Table S1 and the results summarized in Fig. 5. The Cabauw simulations are included in Fig. 5 for direct comparison with the other studies, despite being also plotted in Fig. 4. The average fine particle pH and $\varepsilon(NO_3^-)$ in each study are listed at the top of each plot in Fig. 5 and the plots for



the different studies are arranged with increasing ambient pH from left to right. This order is followed in the following discussion.

Fine particles in the eastern US (SOAS and WINTER studies, Fig. 5a and 5b) are the most acidic among the sites, with average pH of approximately 1 due to the lowest $NH_3$ (and to some minor

extent due to small $NO_3^-$, through its effect on liquid water). In winter (the NE US), $NH_x$ control is most efficient in decreasing $PM_{2.5}$ ion mass since particle pH corresponds to a higher $\varepsilon(NO_3^-)$ (37%) in winter than in summer (22%). $PM_{2.5}$ ion mass reductions from $NO_3^T$ control and $SO_4^{2-}$ control are similar, since aerosol $NO_3^-$ and $SO_4^{2-}$ are comparable in mass. In the southeastern US in summer, $NO_3^T$ control is not effective because $NO_3^-$ only contributed 4% to the $NH_4^+$-$SO_4^{2-}$-

$NO_3^-$ aerosols (Fig. 5a). Because of the small $NO_3^-$ fraction and already low pH in summer, $NH_x$ control only leads to minor reductions in particle ionic mass. In contrast, $SO_4^{2-}$ control produces the highest reduction of particle ionic mass since it is the dominant inorganic species (76%) in this region. Therefore, it is more effective to control $NH_x$ in winter and $SO_4^{2-}$ in summer in the eastern US, a finding consistent with previous studies (Duyzer, 1994; Tsimpidi et al., 2007).

For the southwest US summer (CalNex study, Fig. 5c), since $NO_3^-$ was the most abundant among $NH_4^+$-$SO_4^{2-}$-$NO_3^-$ aerosol components, reducing $NH_x$ is the most effective way to reduce $PM_{2.5}$ ion mass as the ambient particle pH is within the range where $\varepsilon(NO_3^-)$ is sensitive to pH. $NO_3^T$ control follows closely in effectiveness, whereas reducing $SO_4^{2-}$ is the least effective. In the WINTER and CalNex studies, $PM_{2.5}$ ion mass decreases at a lower rate towards higher levels in

$NH_x$ reduction (see Fig. 5b and 5c) due to the nonlinear response in $\varepsilon(NO_3^-)$ to $NH_3$ concentration (as shown in Fig. 3b or Fig. 2). For instance, when $\varepsilon(NO_3^-)$ drops from 50% to 0%, the sensitivities to $NH_3$ keeps decreasing until reaching zero. The pH stays nearly flat for the $NO_3^T$ control and $SO_4^{2-}$ control and decreases with $NH_x$ control.

Cabauw winter and Beijing winter haze conditions (see Fig. 5f and 5e) are similar in terms of

benefits in reducing particle ionic mass from $NH_x$, $NO_3^T$, or $SO_4^{2-}$ controls. This is because of similarities in pH and $\varepsilon(NO_3^-)$ between these sites. For the haze condition in Beijing, $NH_x$ control doesn't produce as much $PM_{2.5}$ ion mass reduction as $NO_3^T$ and $SO_4^{2-}$ controls, unless more than a 60% reduction in $NH_x$ is reached. However, after that PM mass reduction is fast. At 90% $NH_x$ reduction, a decrease of more than half of the particle ionic mass is predicted. $NO_3^T$ and $SO_4^{2-}$

controls produce equivalent results due to the same mass fractions of $NO_3^-$ and $SO_4^{2-}$ (both equal




to 36%) and linear response in particle ionic mass. Comparing the pH profiles, the largest reduction in pH is predicted for Beijing haze if reducing $NH_x$. At 50% $NH_x$ reduction, pH changes from 4.1 to 2.5 in Beijing, whereas pH only changes from 3.9 to 3.3 in Cabauw. This can be explained by differences in $\varepsilon(NH_4^+)$, which is at 60% in Beijing versus 19% in Cabauw.

**3.5 Other implications of lowering pH by $NH_3$ emission control**

The benefit of reducing $NH_3$ emission to reduce ambient $PM_{2.5}$ mass concentrations depends on the conditions at a specific site. While particle pH is lowered during the process, other pH related atmospheric processes are affected. One potentially unintended effect is nitrogen deposition. Nitrogen deposition rates depend on particle versus gas phase fractions, as there is a large

difference between gas and particle deposition velocities. For example, the dry deposition velocity of $NH_3$ is about 1-2 cm s$^{-1}$ over forests, agricultural, or mixed-use land, and 10 times that of $NH_4^+$ (Duyzer, 1994; Schrader and Brummer, 2014). Also, the dry deposition velocity of $HNO_3$ is similar to that of $NH_3$ (Huebert and Robert, 1985). Thus, lowering particle pH produces more localized $HNO_3$ deposition and less localized $NH_3$ deposition near the $NO_x$ and $NH_3$

sources, respectively, since the gases deposit faster than the particle phase.

An addition consequence of lowering particle pH is that it can increase aerosol toxicity. Many studies have identified links between strong particle acidity and adverse health endpoints (Koutrakis et al., 1988; Thurston et al., 1994; Raizenne et al., 1996; Gwynn et al., 2000; Lelieveld et al., 2015). We recently showed one way this can happen is due to increased

conversion of $PM_{2.5}$ insoluble transition metals to soluble forms by strong acidity (Fang et al., 2017), which increases the particles ability to induce oxidative stress (Ghio et al., 2012). Lowering pH may reduce $PM_{2.5}$ mass but increase overall potential for adverse health effects due to significantly greater toxicity of soluble metals relative to ammonium nitrate. Finally, lowering pH can also impact the deposition pattern and bioavailability of trace limiting nutrients such as

Fe, P, and other metals (Meskhidze et al., 2003; Nenes et al., 2011) with important implications for primary productivity (Meskhidze et al., 2005) and even the oxygen state of the subsurface ocean (Ito et al., 2016).





## 4. Summary

In this study, we assess the effectiveness of $NH_3$ control as a way to lower inorganic $PM_{2.5}$ mass based on observational data sets from the US, the Netherlands, and China during different seasons. These sites encompass a diverse range in $NH_3$ and inorganic aerosol concentrations, and

thermodynamic conditions. In all cases, the relative humidities are sufficiently high (average RH > 55%) that a completely deliquesced inorganic phase is a reasonable assumption, which is implicit to the thermodynamic calculations (metastable mode). Focusing on Cabauw, the Netherlands, a site in a region highly impacted by agricultural emissions, and somewhat representative of northwestern Europe, we show that the effectiveness of $NH_3$ control changes with season. In

winter, a much larger reduction in $NH_3$ is required to reduce $NO_3^-$ than in summer, making $NO_x$ control more effective in winter. This is explained by a shift in the $HNO_3$-$NO_3^-$ partitioning ($\varepsilon(NO_3^-)$) curve to lower pH in winter and further from the actual ambient particle pH. A similar situation is seen in Beijing in winter, where $NH_3$ emission control would also be less effective. In most other sites investigated, $NH_3$ control is effective in reducing $PM_{2.5}$ mass, in regions with

reasonably high ammonium nitrate concentrations.

The analysis presented here provides a conceptual and direct evaluation of how the inorganic gas-particle system can be expected to respond to changes in $NH_3$ emissions, and how it contrasts to $NO_x$ control. The approach relies on the single $HNO_3$-$NO_3^-$ partitioning equation and the use of a thermodynamic model to predict pH. Other approaches are also often used to address

this question. Chemical transport models with imbedded thermodynamic sub-modules (such as ISORROPIA) can provide a more detailed analysis that includes other possible impacts of the emission controls, such as ammonia and nitrate deposition and associated environmental impacts. However, the various uncertainties associated with the many simulated processes involved in these models (e.g., emissions and processing) can affect the predicted results and obscure the

fundamental partitioning processes. With the more transparent and accessible approach presented here, this is less an issue. Both approaches have benefits, but whichever analysis is utilized, it is always useful to explicitly report estimated particle pH as it allows assessment of the predictions and provides contrasts between studies at specific sites.

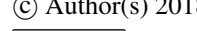



**Acknowledgements**

This work was supported by the National Science Foundation (NSF) under grant AGS-1360730 and by the US Environmental Protection Agency STAR Grant R835882. This publication's contents are solely the responsibility of the grantee and do not necessarily represent the official

views of the US EPA. Further, US EPA does not endorse the purchase of any commercial products or services mentioned in the publication. AN acknowledges support from the European Research Council Project PyroTRACH (Pyrogenic TRansformations Affecting Climate and Health) Grant Agreement 726165.

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





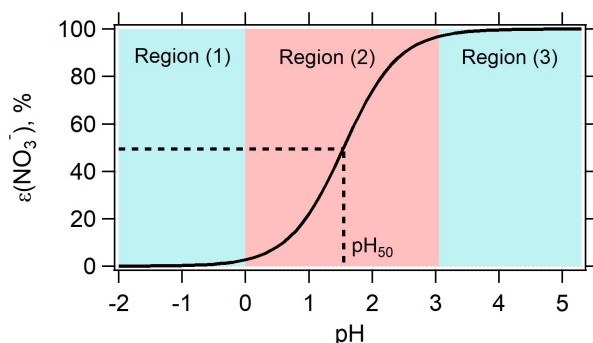

**Figure 1.** Predicted particle phase fraction of total nitrate, $\varepsilon(NO_3^-)$, versus pH for one-year average condition in Cabauw based on Eq. (4). The blue-color zone denotes where $HNO_3$-$NO_3^-$ (nitric acid-nitrate) partitioning is not affected by changes in pH, while the red-color zone shows the region where adjusting pH will change $HNO_3$-$NO_3^-$ partitioning, hence $NO_3^-$ concentration. Greatest sensitivity $NO_3^-$ occurs at $\varepsilon(NO_3^-) = 50\%$, corresponding to $pH_{50}$.





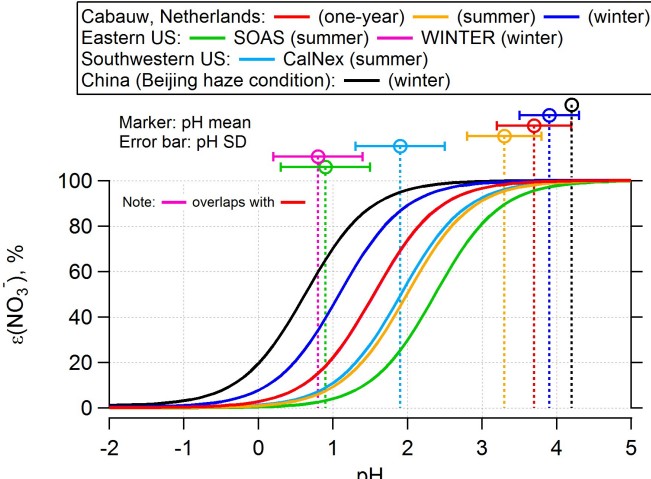

**Figure 2.** $\varepsilon(NO_3^-)$ versus pH for various field studies based on the average temperature, liquid water, and activity coefficients for each study, according to Eq. (4). The WINTER study curve overlaps completely with the Cabauw one-year average curve in red color. The input can be found in Table S1. Vertical lines are the study average ambient fine particle pH calculated with ISORROPIA-II and error bars show the variability in pH as one standard deviation. S-curves and ambient pH for each site or season can be matched by color. For a more direct comparison between seasons at a specific region, supplemental Fig. S4 shows separate curves and ambient pH plots.



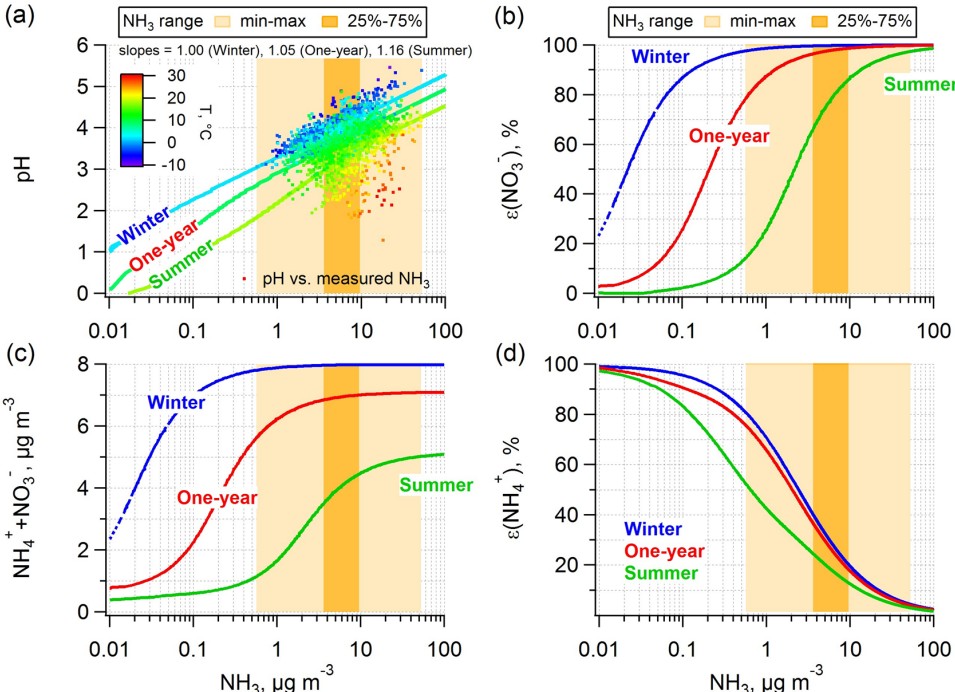

**Figure 3.** Prediction of (a) particle pH, (b) particle phase fractions of total nitrate, $\varepsilon(NO_3^-)$, (c) ammonium and nitrate mass concentration, (d) particle phase fractions of total ammonium, $\varepsilon(NH_4^+)$ for a wide range of ammonia. The simulations are based on the one-year (July 2012-June 2013), summer (June-Aug 2012), and winter (Dec 2012-Feb 2013) average conditions at the Cabauw site with $NH_x$ ($NH_4^+ + NH_3$) left as a free variable. The measured $NH_3$ ranges for the one-year span are also shown as the lighter (min-max) and darker (25%-75% percentiles) orange-color zones. Plot (a) also includes the predicted pH versus 1-hr average measured $NH_3$ data for the entire study and colored by ambient temperature.



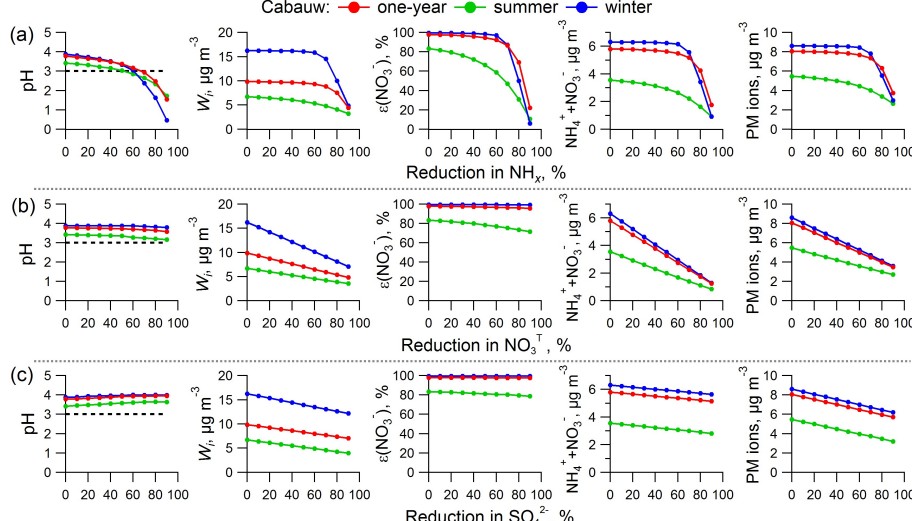

**Figure 4.** ISORROPIA-predicted PM$_{2.5}$ pH (1$^{st}$ column), liquid water content ($W_i$, 2$^{nd}$ column), ε(NO$_3^-$), (3$^{rd}$ column), ammonium and nitrate (4$^{th}$ column), and aerosol inorganic mass

5  concentrations (5$^{th}$ column) as a function of changes in NH$_x$ (NH$_4^+$ + NH$_3$, 1$^{st}$ row), NO$_3^T$ (NO$_3^-$ + HNO$_3$, 2$^{nd}$ row), and SO$_4^{2-}$ (3$^{rd}$ row). Simulations are based on average conditions of one-year, summer, and winter observational data in Cabauw, Netherlands, and changing only NH$_x$, NO$_3^T$ and SO$_4^{2-}$ from the average conditions.



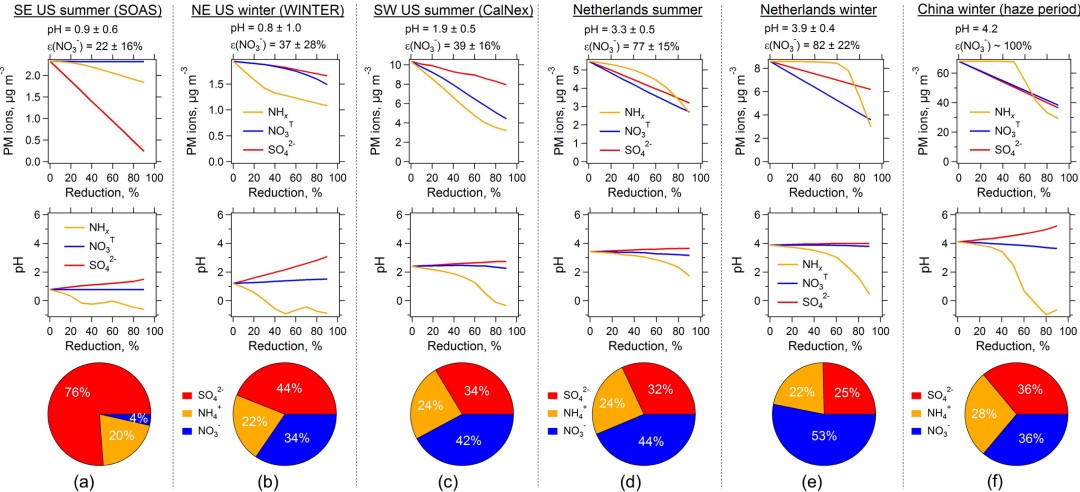

**Figure 5.** Response of predicted $PM_{2.5}$ inorganic mass concentration (1st row) and pH (2nd row)
to reduced levels of $NH_x$ ($NH_3 + NH_4^+$), $NO_3^T$ ($HNO_3 + NO_3^-$), and $SO_4^{2-}$ for several studies
including: (a) the southeastern US summer at a rural ground site in Centreville, AL (SOAS
study), (b) the northeastern US during winter (WINTER aircraft study), (c) the southwestern US
summer at an urban site in Pasadena, CA (CalNex study), (d) & (e) Netherlands summer and
winter conditions at an rural site in Cabauw from this study, and (f) polluted winter conditions
(haze) in Beijing, China. For each case, the average fine ambient particle pH and $\varepsilon(NO_3^-)$, prior
to the reductions, are shown above the figures, with the columns ordered with increasing ambient
particle pH from left to right. $PM_{2.5}$ mass fractions of $NH_4^+$-$SO_4^{2-}$-$NO_3^-$ based on study averages
are shown as pie graphs along the bottom.