# Peer review of "Effectiveness of Ammonia Reduction on Control of Fine Particle Nitrate"

_Atmospheric Chemistry and Physics, 2018_

## Referee Comment (RC1) · Anonymous Referee #1 · 1 Jun 2018

This paper explores the sensitivity of $NH_4NO_3$ concentration to gas phase $NH_3$ and NOx control for a number of contrasting locations and provides a comprehensive evaluation of the effectiveness of ammonia reduction on control of fine particle nitrate. The authors first developed a conceptual basis (S curve) to evaluate the effectiveness of ammonia control on partial nitrate through aerosol pH using a thermodynamic model ISORROPIA-II. Then, they use observation data to calculate the aerosol pH and the S curve for contrasting locations in Netherlands, US and China, and assessed the effectiveness of ammonia reduction in those places. More comprehensive simulations are conducted to investigate the sensitivities of pH and nitrate partitioning to $NH_3$ concentration, as well as the effectiveness of $NH_3$, NOx and $SO_2$ control in reducing fine particle mass for contrasting locations and in different seasons. The authors conclude

that NH3 emissions control would be only be effective in reducing PM2.5 mass when ambient particle pH drops below approximately 3.

The question about how effectiveness is ammonia reduction on air quality amelioration is an area of active research, and the paper adds new results to the literature. In particular, I am very impressed by the conciseness of the S curve and the way it links up the control effectiveness with relating factors. I think this paper fits well into the scope of ACP and will interest its readers. In general, this paper is well-written, and I recommend it to be published in ACP after the following weakness/questions are addressed:

Major comments:

1. Pg 3, line 19: The authors offered clear explanation for the decreasing tendency of SO2 and NOx, which is a result of regulation. However, it seems less clear to me why NH3 is increasing, although the authors have tied NH3 emissions with population growth previously. It would be better to explicitly state that the increase of NH3 emissions is due to the increase of farming activities and fertilizer applications, in order to support the growth of population. I would also suggest adding something about the potential increase of ammonia emission due to global warming, such as the study of Skjøth and Geels 2013. Skjøth, C., and Camilla Geels. "The effect of climate and climate change on ammonia emissions in Europe." Atmospheric Chemistry and Physics 13 (2013): 117-128.

2. Pg 5, line 5. "With high NH3 concentration, it is somewhat representative of northwestern Europe." I would suggest the authors to provide additional evidence for this claim. Perhaps, some reference which indicate that northwestern Europe is normally have high NH3 concentration. Or, maybe provide the averaged NH3 concentration value on northwestern Europe and compared it with the averaged NH3 concentration in Cabauw.

3. Pg 6, line 15. "Inorganic ions are also assumed to be only in the aqueous phase."

Does the model assume that all aerosol species are in the aqueous phase or it also consider some of the species in solid state? Please clarify.

4. Pg 6, line 29. Other studies show that existence of organic phase could also impact the NH3 and NO3 partition as some SOA could react with NH3 and reduce the NH3 concentration. Add comments.

Zhu, S., Horne, J.R., Montoya-Aguilera, J., Hinks, M.L., Nizkorodov, S.A. and Dabdub, D., Modeling reactive ammonia uptake by secondary organic aerosol in CMAQ: application to continental US.

5. Pg 7, line 4-5. The authors used two "discussed below" in this sentence. It would be better to give the exact section or location of the discussion instead. Does it refer to the first paragraph of 2.3?

Actually, there is research showing that different mixing assumption could have significant impact on NO3- and NH4+ partition, especially on NO3-:

Zhu, S., Sartelet, K., Zhang, Y. and Nenes, A., 2016. Three‐dimensional modeling of the mixing state of particles over Greater Paris. Journal of Geophysical Research: Atmospheres, 121(10), pp.5930-5947.

6. P11, line 15. The authors should provide more details regarding to the nature of "particle artifacts in the gas collection system" that is affecting the measurement of HNO3 and HCl.

7. Pg 17, line 19. Since the calculations are based on site measurement in this study, does it suggest that the pH calculated here is closer to the reality than the one calculated by Pozzer et al., (2017). Or, on the other hand, is it possible that the measurements site is not representative enough for the larger domain used in the global model calculation due to its coarse resolution? Are there any regional simulation results that is consistent with the pH prediction presented here?

8. Pg 18, line 13-14. This conclusion looks not very convincing to me. Since the particle

composition is so different between SE US and NE US, the author should justify how the SE US could be a representative case for the eastern US in the summer, and how the NE US could be representative case for the eastern US in the winter before drawing such a conclusion. Or latest explain the cause of such a high sulfate composition (76%) in the SE US case.

Minor Comments:

Pg 5, line 13, the word "alternatively" here is confusing. Do you mean it is the first hour measurement is for PM1 and the next hour will be for PM2.5? In that case the measurement interval will be 2 hours for either PM1 or PM2.5, is that the case? Please clarify.

Pg 7, line 9. "In Cabauw, it has been reported . . ."Could reference be provided for this report?

Pg 7, line 25. It would be better to specify the "coarse mode salts" that HNO3 evolved into.

Pg 9, line 14. "0.987x10-14 is a unit conversion factor" I would better to specify which units are being converted with this factor.

Pg 10, line 3. Could the authors be more specific on how the "approximately 0.6" non-ideality shifts are calculated? Or provide a reference S curve without the non-ideality effect?

P11, line 17. Could the authors provide the references for those "previous studies" mentioned here?

P12, line 2. Could the authors provide the exact hour ranges used in this study to define "night" and "daytime"?

Pg 12, line 13. I found it confusing that the authors keep changing between "NE US" and "WINTER" for the Guo et al., (2016) case, for example, "WINTER" is used in Figure

2, but "NE US" is used here in the text. I suggest the authors use more consistent expression.

Pg 19, line 4. The previous discussions in this paragraph are based on Cabauw winter and Beijing, while the 19% $\varepsilon$(NH4+) value used here are from one-year Cabauw, would you explain why?

Pg 20, line 12. What does "further from the actual ambient particle pH" referred for? Do you mean the region 2 of the curve is further from the ambient particle pH?

---

## Referee Comment (RC2) · Anonymous Referee #2 · 2 Jun 2018

There is growing interest in controlling ammonia emissions to reduce N deposition and NH4NO3. This study focuses on the non-linearities in the response of particulate matter to changes in ammonia concentration. In particular, they introduce a simple conceptual framework (S-curve) to help explain differences in the sensitivity of PM to NH3 emissions at different locations and for different seasons. Such non-linearities have often been overlooked in previous studies. This study is within the scope of ACP. It is well written and the methods are sound. I only have few comments for the authors to address before I can recommend this study for publication.

Comments

1) The authors introduce a new conceptual framework to explain seasonal and regional differences in the sensitivity of particulate matter to ammonia emissions. This has

potential policy implications and it would be useful for the authors to compare with other techniques that have been used previously to highlight potential differences.

In particular, previous studies have used the gas-ratio from Ansari and Pandis to interpret global model results (see for instance Pinder et al. (2007, 20088), Paulot (2016), Pozzer et al., 2017) GR = (TNH4 - 2*TSO4)/TNO3 with 0<GR<1 indicating sensitivity to NHx and GR>1 indicating sensitivity to NH3.

Obviously, this cannot directly address variations associated with seasonality. However, based on the information provided in Table S1, GR∼<1 only for SE US, Virginia, and Pasadena. In other words the weak sensitivity of nitrate to ammonia emissions at the other sites could be inferred simply from concentrations, which is consistent with the findings of the studies mentioned earlier.

In addition, many global models do not use ISORROPIA but simpler (cheaper) aerosol thermodynamic models (see for instance Bellouin et al (2011), Hauglustaine (2014)). Such schemes, which do not explicitly account for aerosol pH, will also simulate a nonlinear response of ammonium nitrate to changes in a ammonia emissions (see equation A8 in Bellouin et al (2011)). It would be useful for the authors to show how different the response of nitrate and ammonium to changes in ammonia/NOx emissions (i.e., Fig 5) would be using such approach.

In particular, this would help strength the case for thinking in terms of aerosol pH rather than simply in terms of concentrations.

2)

I am not convinced by the current discussion of the impact of NH3 emissions controls on nitrogen deposition. The authors argue that lowering aerosol pH (via lower NH3 emissions) will modify the ratio of reduced to oxidized nitrogen deposition. However, it is unclear why this is important (no reference is given), especially considering the benefits of lower NHx deposition and the existence of other removal pathways (wet

deposition) that may not exhibit the same sensitivity to the NH4/NH3 partitioning. A longer discussion is needed given that this conclusion is highlighted in the abstract.

3) the authors focus on seasonal averages. It would be interesting to discuss whether the sensitivity of particulate matter to NH3 emissions is different depending on the concentration of NO3 and whether this would affect the probability distribution of PM under the different emission reduction scenarios shown in Fig. 5. This may be important for policy makers as some standards are based on 24hr averages (https://www3.epa.gov/ttn/naaqs/standards/pm/s_pm_history.html)

Technical comments

4) p4 line 5 NH3 can also enhance the in-cloud oxidation of SO2 by O3. See for instance Wang (2011) or Paulot (2017)

5) p17 line 15 I believe livestock emissions are likely to dominate ammonia emissions in summer.

6) dash black line Fig. 4 not defined

Wang, S., J. Xing, C. Jang, Y. Zhu, J. S. Fu, and J. Hao (2011), Impact assessment of ammonia emissions on inorganic aerosols in east China using response surface modeling technique, Environ. Sci. Technol., 45, 9293–9300.

Paulot, F., S. Fan, and L. W. Horowitz (2017), Contrasting seasonal responses of sulfate aerosols to declining SO2 emissions in the Eastern U.S.: Implications for the eflcacy of SO2 emission controls, Geophys. Res. Lett., 44, 455–464, doi:10.1002/2016GL070695.

Ansari, A. S. and Pandis, S. N.: Response of Inorganic PM to Pre- cursor Concentrations, Environ. Sci. Technol., 32, 2706–2714, 1998.

Hauglustaine, D. A.; Balkanski, Y. & Schulz, M. A global model simulation of present and future nitrate aerosols and their direct radiative forcing of climate Atmos. Chem.

Phys., 2014, 14, 11031-11063

Pinder et al. Observable indicators of the sensitivity of PM2.5 nitrate to emission reductions—Part I: Derivation of the adjusted gas ratio and applicability at regulatory-relevant time scales Atmospheric Environment 42 (2008) 1275–1286

Pinder, R. W.; Gilliland, A. B. & Dennis, R. L. Environmental impact of atmospheric NH3 emissions under present and future conditions in the eastern United States Geophys. Res. Lett., 2008, 35

Pinder, R. W.; Adams, P. J. & Pandis, S. N. Ammonia emission controls as a cost-effective strategy for reducing atmospheric particulate matter in the Eastern United States Environ. Sci. Technol., 2007, 41, 380-386

Paulot, F., Ginoux, P., Cooke, W. F., Donner, L. J., Fan, S., Lin, M.-Y., Mao, J., Naik, V., and Horowitz, L. W.: Sensitivity of nitrate aerosols to ammonia emissions and to nitrate chemistry: implications for present and future nitrate optical depth, Atmos. Chem. Phys., 16, 1459-1477, https://doi.org/10.5194/acp-16-1459-2016, 2016.

Aerosol forcing in the Climate Model Intercomparison Project (CMIP5) simulations by HadGEM2‐ES and the role of ammonium nitrate Nicolas Bellouin; Jamie Rae; Andy Jones; Colin Johnson; Jim Haywood; Olivier Boucher Journal of Geophysical Research: Atmospheres (1984–2012). 2011, DOI: 10.1029/2011JD016074

---

## Author Comment (AC1) · 2 Aug 2018

We thank the referees for their thoughtful and constructive comments. We have addressed the comments (numbered, below), with referee comments in quotes and italics, and our responses in plain text.

**Referee #1**

Major comments:

1. *"Pg 3, line 19: The authors offered clear explanation for the decreasing tendency of SO2 and NOx, which is a result of regulation. However, it seems less clear to me why NH3 is increasing, although the authors have tied NH3 emissions with population growth previously. It would be better to explicitly state that the increase of NH3 emissions is due to the increase of farming activities and fertilizer applications, in order to support the growth of population. I would also suggest adding something about the potential increase of ammonia emission due to global warming, such as the study of Skjøth and Geels 2013. Skjøth, C., and Camilla Geels. "The effect of climate and climate change on ammonia emissions in Europe." Atmospheric Chemistry and Physics 13 (2013): 117-128."*

   To emphasize the relationship between $NH_3$ emission and population due to food production, we have revised Page 3 Line 11 to "Given that fertilizer usage supports food production for about half the global population (Erisman et al., 2008), $NH_3$ emissions are linked to world population and so expected to increase into the 21th century (Gerland et al., 2014)." In the introduction, we have added a sentence citing the suggested publication, "Higher temperatures resulting from global warming can also potentially enhance $NH_3$ emissions (Skjøth and Geels, 2013)."

2. *"Pg 5, line 5. "With high NH3 concentration, it is somewhat representative of northwestern Europe." I would suggest the authors to provide additional evidence for this claim. Perhaps, some reference which indicate that northwestern Europe is normally have high NH3 concentration. Or, maybe provide the averaged NH3 concentration value on northwestern Europe and compared it with the averaged NH3 concentration in Cabauw."*

   We have provided more information and noted that Cabauw has high $NH_3$ levels; "Northwestern Europe has fairly high $NH_3$ concentrations with yearly averages ranging from 1 to 14 µg m$^{-3}$ (median as 4.2 µg m$^{-3}$) for the Netherlands in 2013, reported by the Measuring Ammonia in Nature (MAN) network (Lolkema et al., 2015). Satellite-derived 14 years average for the western Europe is 3 ppbv (~2.3 µg m$^{-3}$) (Warner et al., 2017). Cabauw was somewhat higher due to intensive agriculture in the region with observed yearly $NH_3$ average of $7.3 \pm 6.0$ µg m$^{-3}$ (~10 ppbv)."

3. *"Pg 6, line 15. "Inorganic ions are also assumed to be only in the aqueous phase." Does the model assume that all aerosol species are in the aqueous phase or it also consider some of the species in solid state? Please clarify."*

Yes, we ran the model assuming that all ions are in the aqueous phase. We have revised as "Inorganic ions are also assumed to be only in the aqueous phase (i.e., no solid precipitates)." to minimize confusion.

4. *"Pg 6, line 29. Other studies show that existence of organic phase could also impact the NH3 and NO3 partition as some SOA could react with NH3 and reduce the NH3 concentration. Add comments. Zhu, S., Horne, J.R., Montoya-Aguilera, J., Hinks, M.L., Nizkorodov, S.A. and Dabdub, D., Modeling reactive ammonia uptake by secondary organic aerosol in CMAQ: application to continental US."*

We have revised to "This is confirmed by the good agreement between measured and ISORROPIA-II predicted $NH_3$-$NH_4^+$ partitioning without considering organic acids or other organic species (see section 3.2). Although recent modeling study has suggested that ambient $NH_3$ concentration can be decreased by as much as 31% in winter and 67% in summer in the US, due to the reactive uptake of $NH_3$ by secondary carbonyl compounds (Zhu et al., 2018), this process doesn't appear to have an impact on $NH_3$-$NH_4^+$ partitioning and predicted pH for the locations in this study."

With the above said, it is also important to note that even if $NH_{3(g)}$ were reduced by 30-60% by reactions with the organic phase, the impact on aerosol acidity would be modest (change of about 0.1-0.2 pH units) given that an order of magnitude change in ambient $NH_3$ concentration is required for pH levels to be changed by one unit (Guo et al., 2017).

5. *"Pg 7, line 4-5. The authors used two "discussed below" in this sentence. It would be better to give the exact section or location of the discussion instead. Does it refer to the first paragraph of 2.3? Actually, there is research showing that different mixing assumption could have significant impact on NO3- and NH4+ partition, especially on NO3-: Zhu, S., Sartelet, K., Zhang, Y. and Nenes, A., 2016. Threeˆ˘ARˇ dimensional modeling of the mixing state of particles over Greater Paris. Journal of Geophysical Research: Atmospheres, 121(10), pp.5930-5947."*

We have provided the sections that we refer to, "(discussed below in Section 2.3)…(discussed below in Section 3.2 and also see Table S1)". Thanks for pointing out this paper, we have cited it in the manuscript. We have not discussed it in detail since it follows the less quantitative approach of assuming nitrate only forms once sulfate is fully neutralized (i.e., $NH_4^+/SO_4^{2-}$ ratios above 2), instead of a rigorous thermodynamic analysis, the focus of this paper.

6. *"P11, line 15. The authors should provide more details regarding to the nature of "particle artifacts in the gas collection system" that is affecting the measurement of HNO3 and HCl."*

We have edited this, removing speculation on the cause. It new reads, "However, for unknown reasons, gas-phase components of these two species showed significant discrepancies ($R^2$ of 0.13 to 0.17). We note that it may be associated with the very low gas phase concentrations of these species, in contrast to $NH_3$."

7. *"Pg 17, line 19. Since the calculations are based on site measurement in this study, does it suggest that the pH calculated here is closer to the reality than the one calculated by Pozzer et al., (2017). Or, on the other hand, is it possible that the measurements site is not representative enough for the larger domain used in the global model calculation due to its coarse resolution? Are there any regional simulation results that is consistent with the pH prediction presented here?"*

This is a good point. Pozzer et al. (2017) did not publish any $NH_3$ concentration, which could be used for comparison with observations in Cabauw. Other relevant regional simulated pH could also not be found for the same region. Pozzer et al. (2017) published an average pH for Europe of ~2, which is 1.7 units lower than the one-year pH in Cabauw. This may indicate that the Cabauw sampling site is not representative of Europe in general (the Pozzer's paper), but a detailed comparison between the two model inputs is necessary to understand the cause of the pH difference. As noted above (and in the revised manuscript), the Cabauw site has higher $NH_3$ concentrations, which will increase particle pH. Holding all other model inputs constant, a factor of 10 lower $NH_3$ decreases pH by roughly one unit (as shown in Figure 3). Based on satellite derived 14-year $NH_3$ average (3 ppb and 2.3 µg m$^{-3}$) for western Europe as an example (Warner et al., 2017), the pH in Cabauw decrease to around 3 for the one-year average and winter average, and below 3 for summer, according to the linear fitting lines in Figure 3. Meteorological conditions and particle composition also contribute to the difference in pH prediction mentioned above. Although it has been found that inaccurate treatment of nonvolatile cations may cause overestimation of particle pH in regional models (Vasilakos et al., 2018), it doesn't explain the above pH difference since the modeled pH in Pozzer et al. (2017) (from global modeling) is the lower one and the levels of nonvolatile cations are low in Cabauw. Despite of the difference in pH, we believe the findings of Pozzer et al, (2017) are consistent with the framework established in this study, that is a critical pH of ~3. Since the Pozzer's European pH is 2, below 3, controlling $NH_3$ emission is suggested as an effective way to reduce particle mass.

8. *"Pg 18, line 13-14. This conclusion looks not very convincing to me. Since the particle composition is so different between SE US and NE US, the author should justify how the SE US could be a representative case for the eastern US in the summer, and how the NE US could be representative case for the eastern US in the winter before drawing such a conclusion. Or latest explain the cause of such a high sulfate composition (76%) in the SE US case."*

We have clarified the statement as suggested by the reviewer. The SE US simulation is only representative of the SE US; the same for the NE US simulation. Now the text becomes "Therefore, it is more effective to control $NH_x$ in winter in the NE US and $SO_4^{2-}$ in summer in SE US, a finding consistent with previous studies (Duyzer, 1994; Tsimpidi et al., 2007)." The large fraction of sulfate is a result of the small fraction of nitrate. In such situations, ammonium basically tracks sulfate. Due to the difference in molecular weight, sulfate is the dominant inorganic mass. We have added a sentence explaining the reason, "A small fraction of nitrate aerosol is typically observed in the southeast in summer (Hidy et al., 2014) due to the high temperature and low particle pH."

Minor comments:

9. *"Pg 5, line 13, the word "alternatively" here is confusing. Do you mean it is the first hour measurement is for PM1 and the next hour will be for PM2.5? In that case the measurement interval will be 2 hours for either PM1 or PM2.5, is that the case? Please clarify."*

A clarification is made. "…alternatively between $PM_1$ and $PM_{2.5}$, each size sampled hourly (i.e., a two-hour interval for one size; a one-hour interval for gas)."

10. *"Pg 7, line 9. "In Cabauw, it has been reported…" Could reference be provided for this report?"*

The reference "(Schlag et al., 2017)" was there in the middle of the sentence. Since it is not obvious, we have moved it to the end.

11. *"Pg 7, line 25. It would be better to specify the "coarse mode salts" that HNO3 evolved into."*

We have added examples as "coarse mode salts (e.g., NaCl and $CaCl_2$)".

12. *"Pg 9, line 14. "0.987x10-14 is a unit conversion factor" I would better to specify which units are being converted with this factor."*

We have added explanation as "where $0.987 \times 10^{-14}$ is a unit conversion factor (from converting atm and µg to SI units)".

13. *"Pg 10, line 3. Could the authors be more specific on how the "approximately 0.6" nonideality shifts are calculated? Or provide a reference S curve without the non-ideality effect?"*

The 0.6 unit pH difference is provided by comparing nitrate partitioning S curves calculated assuming $\gamma_{NO_3^-}\gamma_{H^+} = 1$ (ideal) and 0.24 (non-ideal; from ISORROPIA). More specifically, compare $pH_{50}$ values for the two S curves. A figure is provided to visualize the difference and added to supplemental material as the new Fig. S2.

[Figure]

Figure. Predicted particle phase fraction of total nitrate, $\varepsilon(NO_3^-)$, versus pH for one-year average condition in Cabauw based on Eq. (4). The red and blue lines are based on $\gamma_{NO_3^-}\gamma_{H^+} = 0.24$ and 1, respectively.

14. *"P11, line 17. Could the authors provide the references for those "previous studies" mentioned here?"*

A reference has been added.

15. *"P12, line 2. Could the authors provide the exact hour ranges used in this study to define "night" and "daytime"?"*

We define "night" and "day" by sunrise and sunset. However, we don't have solar radiation data to plot a diurnal profile. Since sunrise and sunset time can vary substantially from summer to winter, we cannot provide exact hour ranges. For example, daytime is from 05:24 to 22:03 on June 1 2013 and from 08:48 to 16:38 on Dec 31 2013. To minimize confusion, we have revised the text to exact hours which are not as affected by seasonal changes in sunrise and sunset, "A diurnal pattern of ambient particle pH is observed in Cabauw, similar to other studies (Guo et al., 2015). For example, for the nighttime period of 1 am to 7 am, the average pH is 3.9, whereas for the daytime period of 1 pm to 6 pm the pH is 3.5. The difference is mainly driven by the diurnal variation in liquid water content (see Fig. S1)".

16. *"Pg 12, line 13. I found it confusing that the authors keep changing between "NE US" and "WINTER" for the Guo et al., (2016) case, for example, "WINTER" is used in Figure 2, but "NE US" is used here in the text. I suggest the authors use more consistent expression."*

We have revised the Figure 2 legends to separate the SOAS and WINTER studies. The SOAS study is under the "Southeastern US" and the WINTER study is under the "Northeastern US". Hopefully, this clarifies the issue.

17. *"Pg 19, line 4. The previous discussions in this paragraph are based on Cabauw winter and Beijing, while the 19% "(NH4+) value used here are from one-year Cabauw, would you explain why?"*

Thanks for pointing it out! We have replaced "19%" with "27%", which was the right number for Cabauw winter average.

18. *"Pg 20, line 12. What does "further from the actual ambient particle pH" referred for? Do you mean the region 2 of the curve is further from the ambient particle pH?"*

Yes, we mean region 2 or $pH_{50}$. We have revised to "This is explained by a shift in the HNO$_3$-NO$_3^-$ partitioning ($\varepsilon(NO_3^-)$) curve to lower pH in winter and $pH_{50}$ (where $\varepsilon(NO_3^-) = 50\%$) further from the actual ambient particle pH.".

**References**

Duyzer, J.: Dry deposition of ammonia and ammonium aerosols over heathland, J. Geophys. Res., 99, 18757-18763, doi: 10.1029/94jd01210, 1994.

Guo, H., Xu, L., Bougiatioti, A., Cerully, K. M., Capps, S. L., Hite, J. R., . . . Weber, R. J.: Fine-particle water and pH in the southeastern United States, Atmos. Chem. Phys., 15, 5211-5228, doi: 10.5194/acp-15-5211-2015, 2015.

Guo, H., Weber, R. J., and Nenes, A.: High levels of ammonia do not raise fine particle pH sufficiently to yield nitrogen oxide-dominated sulfate production, Sci. Rep., 7, doi: 10.1038/s41598-017-11704-0, 2017.

Hidy, G. M., Blanchard, C. L., Baumann, K., Edgerton, E., Tanenbaum, S., Shaw, S., . . . Walters, J.: Chemical climatology of the southeastern United States, 1999-2013, Atmos. Chem. Phys., 14, 11893-11914, doi: 10.5194/acp-14-11893-2014, 2014.

Lolkema, D. E., Noordijk, H., Stolk, A. P., Hoogerbrugge, R., van Zanten, M. C., and van Pul, W. A. J.: The Measuring Ammonia in Nature (MAN) network in the Netherlands, Biogeosciences, 12, 5133-5142, doi: 10.5194/bg-12-5133-2015, 2015.

Pozzer, A., Tsimpidi, A. P., Karydis, V. A., de Meij, A., and Lelieveld, J.: Impact of agricultural emission reductions on fine-particulate matter and public health, Atmos. Chem. Phys., 17, 12813-12826, doi: 10.5194/acp-17-12813-2017, 2017.

Skjøth, C. A., and Geels, C.: The effect of climate and climate change on ammonia emissions in Europe, Atmos. Chem. Phys., 13, 117-128, doi: 10.5194/acp-13-117-2013, 2013.

Tsimpidi, A. P., Karydis, V. A., and Pandis, S. N.: Response of Inorganic Fine Particulate Matter to Emission Changes of Sulfur Dioxide and Ammonia: The Eastern United States as a Case Study, J Air Waste Manan. Assoc., 57, 1489-1498, doi: 10.3155/1047-3289.57.12.1489, 2007.

Vasilakos, P., Russell, A., Weber, R., and Nenes, A.: Understanding nitrate formation in a world with less sulfate, Atmos. Chem. Phys. Disc., 1-27, doi: 10.5194/acp-2018-406, 2018.

Warner, J. X., Dickerson, R. R., Wei, Z., Strow, L. L., Wang, Y., and Liang, Q.: Increased atmospheric ammonia over the world's major agricultural areas detected from space, Geophys. Res. Lett., doi: 10.1002/2016gl072305, 2017.

Zhu, S., Horne, J. R., Montoya-Aguilera, J., Hinks, M. L., Nizkorodov, S. A., and Dabdub, D.: Modeling reactive ammonia uptake by secondary organic aerosol in CMAQ: application to the continental US, Atmos. Chem. Phys., 18, 3641-3657, doi: 10.5194/acp-18-3641-2018, 2018.

---

## Author Comment (AC2) · 2 Aug 2018

We thank the referees for their thoughtful and constructive comments. We have addressed the comments (numbered, below), with referee comments in quotes and italics, and our responses in plain text.

**Referee #2**

Major comments:

1. *"The authors introduce a new conceptual framework to explain seasonal and regional differences in the sensitivity of particulate matter to ammonia emissions. This has potential policy implications and it would be useful for the authors to compare with other techniques that have been used previously to highlight potential differences.*

   *In particular, previous studies have used the gas-ratio from Ansari and Pandis to interpret global model results (see for instance Pinder et al. (2007, 2008), Paulot (2016), Pozzer et al., 2017) GR = (TNH4 - 2\*TSO4)/TNO3 with 0<GR<1 indicating sensitivity to NHx and GR>1 indicating sensitivity to NH3.*

   *Obviously, this cannot directly address variations associated with seasonality. However, based on the information provided in Table S1, GR_<1 only for SE US, Virginia, and Pasadena. In other words the weak sensitivity of nitrate to ammonia emissions at the other sites could be inferred simply from concentrations, which is consistent with the findings of the studies mentioned earlier.*

   *In addition, many global models do not use ISORROPIA but simpler (cheaper) aerosol thermodynamic models (see for instance Bellouin et al (2011), Hauglustaine (2014)). Such schemes, which do not explicitly account for aerosol pH, will also simulate a nonlinear response of ammonium nitrate to changes in a ammonia emissions (see equation A8 in Bellouin et al (2011)). It would be useful for the authors to show how different the response of nitrate and ammonium to changes in ammonia/NOx emissions (i.e., Fig 5) would be using such approach.*

   *In particular, this would help strength the case for thinking in terms of aerosol pH rather than simply in terms of concentrations."*

   The reviewer raises an important and very broad question. First we note that global models use aerosol thermodynamic modules of all levels of complexity (some not at all). All these models would predict some degree of nonlinearity because one of the precursors, $NH_3$ or $HNO_3$, become limiting. Our point is that using pH to look at the sensitivity of nitrate to the precursors is new. It makes things simpler and provides a more fundamental understanding of the processes involved. Furthermore, even if the models have the correct thermodynamics they can still get the sensitivity wrong due to a biased predicted pH, as we note with the reference to Vasilakos et al. (2018). We feel that the degree to which each implementation differs, and how it compares with the usage of pH as a control parameter requires a dedicated publication in itself.

   We have added some text to try and clarify these points. The next now reads:

"Large-scale models to assess effectiveness of $NH_3$ control requires good predictions of a range of pertinent emissions and sinks ($NH_3$, $NO_x$, $SO_2$, and nonvolatile cations), and accurate representation of their applicable atmospheric chemical processes. Thermodynamic modules of different levels of complexity are then applied to determine sensitivities to the precursors (e.g., $NH_3$, $HNO_3$). In some cases (Pozzer et al., 2017), the aerosol pH is explicitly determined with an embedded thermodynamic model, such as ISORROPIA-II (Fountoukis and Nenes, 2007). Due to the complexities from all these factors, chemical transport model-predicted responses to changing emissions may not align with observations. For example, the sensitivity of $PM_{2.5}$ pH in the Community Multiscale Air Quality (CMAQ) simulations to the mass of crustal material apportioned to the $PM_{2.5}$ size range can have important effects on anticipated responses to these changing emission trends. Vasilakos et al. (2018) have shown that including too much crustal material in $PM_{2.5}$ results in a predicted increasing trend in both aerosol pH and concentrations of $NH_4NO_3$, which is counter to observations (Weber et al., 2016).

Overall, calculating aerosol pH is a more accurate approach that provides a fundamental understanding of the factors controlling $HNO_3$-$NO_3^-$ partitioning and therefore enables a direct evaluation of different studies. Furthermore, it is also useful to determine aerosol pH since it has broad application to many other important aerosol processes. For instance, pH is a mediator of many heterogeneous chemical processes, including various acid-catalyzed reactions (Jang et al., 2002; Eddingsaas et al., 2010; Surratt et al., 2010), gas-particle partitioning of species other than $HNO_3$ and $NH_3$, such as organic acids and halogens (Fridlind and Jacobson, 2000; Young et al., 2013; Guo et al., 2017; Nah et al., 2018), and solubility of metals and other nutrient species (Meskhidze et al., 2003; Nenes et al., 2011; Longo et al., 2016; Stockdale et al., 2016; Fang et al., 2017).

In this study, we apply a more direct approach, where measured gas and particle concentrations and the thermodynamic model ISORROPIA-II are used directly in a sensitivity analysis to evaluate the effectiveness of $NH_3$ emission controls on fine particle mass relative to $NO_x$ control. Contrasts are made between sites that have a wide range in $NH_3$ concentrations and aerosol composition, …"

2. *"I am not convinced by the current discussion of the impact of NH3 emissions controls on nitrogen deposition. The authors argue that lowering aerosol pH (via lower NH3 emissions) will modify the ratio of reduced to oxidized nitrogen deposition. However, it is unclear why this is important (no reference is given), especially considering the benefits of lower NHx deposition and the existence of other removal pathways (wet deposition) that may not exhibit the same sensitivity to the NH4/NH3 partitioning. A longer discussion is needed given that this conclusion is highlighted in the abstract."*

We were only focusing here on effects on dry deposition since the paper discusses relative gas and particle concentrations and we note the large differences in gas/particle deposition velocities. Discussing effects of N deposition due to wet processing is beyond the scope of this paper. However, the reviewer's point that it may be more complicated is well taken. We have modified the text to be more precise and note complicating effects of wet removal processes.

In the abstract, it has been revised to "Finally, controlling $NH_3$ emissions to increase aerosol acidity and evaporate $NH_4NO_3$ will have other effects, beyond reduction of $PM_{2.5}$ $NH_4NO_3$,

such as increasing aerosol toxicity and potentially altering the deposition patterns of nitrogen and trace nutrients."

In the section 3.5, "Lowering particle pH through $NH_3$ reductions will decrease overall reduced nitrogen deposition but may results in more localized oxidized nitrogen dry deposition if the lower pH results in $NO_3^-$ evaporation and higher $HNO_3$ concentrations. Deposition due to wet removal processes are not considered here."

3. *"the authors focus on seasonal averages. It would be interesting to discuss whether the sensitivity of particulate matter to NH3 emissions is different depending on the concentration of NO3 and whether this would affect the probability distribution of PM under the different emission reduction scenarios shown in Fig. 5. This may be important for policy makers as some standards are based on 24hr averages (https://www3.epa.gov/ttn/naaqs/standards/pm/s_pm_history.html)"*

We understand the point raised. The main issue with this suggestion is that the sensitivity of particulate matter to $NH_3$ (or $HNO_3$) emissions is directly determined by the aerosol pH. Otherwise, the sensitivity can take a wide range of values for constant $NO_3^-$, as the pH can still vary considerably. We believe that this shift in approach (first looking at pH and then seeing how that affects aerosol sensitivity to emissions), is one of the most important messages of the paper. Towards that, a simpler approach, the $HNO_3$-$NO_3^-$ S curve (in Section 2.3), is provided to roughly estimate the effectiveness of $NH_3$ control.

Technical comments:

4. *"p4 line 5 NH3 can also enhance the in-cloud oxidation of SO2 by O3. See for instance Wang (2011) or Paulot (2017)"*

Thanks for bringing attention to these references. We have revised the text to "Reduction in $NH_3$ also reduces the amount of $NH_4^+$ associated with sulfates and lowers the pH-dependent sulfate production rate, such as in cloud $SO_2$ oxidation by $O_3$ (Wang et al., 2011; Cheng et al., 2016; Paulot et al., 2017), and the interplay between the two species may drive much of the sensitivity of $PM_{2.5}$ to $NH_3$ and $NO_x$ reductions (e.g., (Vasilakos et al., 2018))".

5. *"p17 line 15 I believe livestock emissions are likely to dominate ammonia emissions in summer."*

We had thought that as well, but literature studies do not seem to agree. Based on Figure 7 in Zhang et al. (2018), livestock waste dominates in winter rather than summer. The annual emissions from fertilizer and livestock waste are quite similar (5.05 vs 5.31 Tg $a^{-1}$).

6. *"dash black line Fig. 4 not defined"*

We apologize for this oversight. The black dash lines in the pH figures identifies the critical pH value of 3, and now has been noted in the caption.

**References**

[revised manuscript text omitted]